# Hierarchical Structure Model of Safety Risk Factors in New Coastal Towns: A Systematic Analysis Using the DEMATEL-ISM-SNA Method

**DOI:** 10.3390/ijerph191710496

**Published:** 2022-08-23

**Authors:** Chenlei Guan, Damin Dong, Feng Shen, Xin Gao, Linyan Chen

**Affiliations:** 1School of Economics and Management, Tongji University, Shanghai 200092, China; 2School of Business, East China University of Science and Technology, Shanghai 200237, China; 3Shanghai Tongji Engineering Cousulting Co., Ltd., Shanghai 200092, China; 4Department of Building and Real Estate, The Hong Kong Polytechnic University, Hong Kong, China

**Keywords:** safety risk, urbanization, coastal town, public health, sustainable development, DEMATEL-ISM-SNA method, hierarchical structure model

## Abstract

When a coastal town transforms from a rural area to an emerging city, it faces many safety risks. Some are new risks from urban construction, while some are traditional risks that belong to this coastal area. The joint efforts of these risks may lead to new hazards, harming public health, but this problem has not been noticed in previous studies. Therefore, this study constructs the Triangular Framework for Safety Risk in New Towns to identify the risks and proposes strategies to reduce the risks. In this study, multiple methods are integrated, including Decision-Making Trial and Evaluation Laboratory (DEMATEL), Interpretive Structural Modeling (ISM), and Social Network Analysis (SNA). This study takes the Lin-gang Special Area in China as a case study to verify the framework’s effectiveness. Sixteen disaster-causing factors are identified, and the internal linkages among these factors are clarified. Results show that the hybrid method performs well in quantitatively analyzing the risk factors of new coastal towns. A typhoon, public risk perception, and population migration are essential influencing factors. Disaster prevention capability of high-rise buildings, disaster prevention capacity of port facilities, and transportation are the most direct influencing factors. Environmental degradation is the most conductive among all elements. This study contributes to the theoretical theory by proposing an effective framework to analyze the safety risks in new coastal towns. In addition, it provides practical references for governments to make emergency plans in the city.

## 1. Introduction

Safety is the most basic human survival requirement and the top priority for urban construction [1]. Urbanization is an unavoidable construction trend in social development, but it brings many pressures to city safety. In China, new urbanized cities have become vulnerable to continuous challenges [2,3]. For coastal towns, typhoons, storm surges, and other disasters related to natural hazards have become more common, threatening city safety. Numerous components and their interconnection expose cities to many disasters, harming public health. However, the overlap of traditional and emerging threats is difficult to grasp in a new area. Therefore, understanding how to enhance disaster preventive capacities of new coastal cities is an urgent task.

Over the past two decades, studies have provided important information on risk interdependence among critical infrastructures by disasters related to natural hazards and the growing complexity of towns. When a new coastal village transforms from a rural area to an emerging city, population migration, infrastructure construction, and spatial layout are all part of its planning, construction, and operation. It will inevitably lead to overlapping traditional and emerging risks, making the city governance more complex than its counterparts [4]. Urban planning is an effective measure through long-term thinking and decision making to guide future action [5]. It does have a special section on disaster prevention and mitigation. However, many plans are not systematic and targeted enough to avoid risks [6]. Therefore, we carry out comprehensive disaster prevention and mitigation planning research. The aim is to identify the risk coupling in building new coastal towns and break this coupling effect. We hope to provide practical suggestions for new coastal town construction.

Although many studies on coastal urbanization, few studies develop specific methods and tools to capture safety. Previous research conducted resilience arrangements analysis to establish different risk responses [7,8]. Researchers applied some approaches to risk management (e.g., Social Network Analysis (SNA) [9], Decision-Making Trial and Evaluation Laboratory (DEMATEL) [10], Interpretive Structural Modeling (ISM) [11], Bayesian belief network [12]). Such approaches still fail to address the problem systematically due to their shortcomings and advantages [13]. Considering their characteristics in dealing with safety risks, some of these methods provide complementary perspectives [14]. Still, they have not been integrated to identify and resolve the complex linkages that threaten public security in China’s rapidly urbanizing cities. To solve this problem, this study constructs the triangular framework for safety risk in new coastal towns and combines the DEMATEL, the ISM, and the SNA to comprehend the risk characteristics of the city in China, particularly a new coastline zone. It investigates the critical elements and their cascading effects. This study can provide theoretical references for making emergency plans and blocking risks.

## 2. Literature Review

### 2.1. Safety Risk

According to ISO 31000: 2018—Risk management—Guidelines, the risk is the effect of uncertainty on objectives, which can be positive, negative, or both [15]. It is necessary to avoid adverse risks in advance [16]. YoungCheon et al. demonstrated that the safety risk is an antecedent factor for other risk factors [17]. The concept of risk society was first proposed by Baker, who believed that risk society is an inevitable product of modernization [18]. The problems encountered by urbanization are focused on the safety risks. According to Ullah et al., safety risks are undesirable situations, mainly caused by climate change and environmental hazards. They are harmful to social sustainability [19]. This paper defines *safety risk* as the adverse factors affecting cities’ construction and industrial operation.

There has been extensive research regarding urban safety risks. Urban safety issues stem from the risks and uncertainties [20]. Since its formation, it has been subject to various disturbances both within and beyond, including natural and manufactured calamities [21]. Giles-Corti et al. found that urban safety risks are multifactorial and do not exist in isolation [22]. The complexity of exposure may influence the extent to which an area addresses its risks [23].

### 2.2. Triangular Framework for Pubilc Safety and Technology

There are several frameworks, models, and theories. Such as the framework for managing a public safety risk network [3], security, resilience, and sustainability (SRS) [24], the triangular framework for public safety and technology (PST) [25], comprehensive screening, key analysis, and comprehensive evaluation (CKC) [26] and others that have been utilized for managing urban safety risk. The triangular public safety and technology framework is widely accepted in public safety. Fan et al. creatively proposed this model to reduce the impact of emergencies on human society, which uses three edges to represent emergencies, disaster carriers, and emergency management [25]. Zhong et al. constructed a framework for public safety weather hazard systems based on the public safety model proposed by Fan with an additional component: hazard-inducing environments (HIEs) [27]. Zhou et al. incorporated the “human” factor into the public safety science and technology model and concluded that people are both disaster-causing and disaster-carrying vehicles. So they proposed a population-driven social governance strategy from the management dimension [28].

Research on causal factors, especially natural hazards, has transitioned from single to multiple threats, mainly in: (1) multi-hazard correlation studies, e.g., to capture interactions in complex urban systems, Depietri et al. studied heat waves, inland flooding, and coastal flooding and proposed an improved metropolitan multi-hazard risk assessment method [29]; (2) multi-hazard risk analysis, e.g., Han et al. presented an improved method for urban multi-hazard risk assessment [30]. In addition to natural hazard impacts, recent literature has reinforced concerns about the possibility of land subsidence due to urbanization, which may lead to geologic hazards [22]. Urban expansion into nature brings improved recreational activities but increases the conflict between local buildings and fire exposure [31]; a study by Johnson showed that population density affects the combined risk of cities [32].

As indicated in Figure 1, the triangle public safety science and technology model is applied to the safety risk of new coastal towns, including disaster-causing factors, disaster-bearing systems, and solutions. Only by coordinating these parts, formulating plans, and implementing them can comprehensively reduce safety risks [25]. 

The disaster-causing factors are the factors that bring risks to the construction and development of new towns. This study summarizes four categories for safety in new towns based on previous research [25,27,33,34].
Natural disasters (S1): Natural disasters refer to disasters related to natural hazards which occur or are likely to happen in this area and cause heavy losses.Safety accidents (S2): Safety accidents caused by the industry or bring risks to the industry.Urban construction (S3): Urban construction comes with long-term site safety risks, population migration, risks posed by buildings, and other related risks.Public events (S4): Public events refer to events widely concerned or followed up by social groups at different levels.

Research has found disaster-bearing bodies, such as individuals [28,35] and urban lifeline systems [36,37], are also disaster-causing variables. They are divided into parks and communities based on their functional orientation. Additionally, in terms of existence, they are split into physical urban entities and the financial information society related to people. 

The solutions are taken to cope with risks, guaranteeing the stable development of new towns. They include technological and humanistic measures in terms of content, while conventional prevention and emergency measures are in terms of urgency. Safety risks are present throughout, including risk identification, assessment, monitoring, and controlling [38].

### 2.3. DEMATEL, ISM, SNA, and Their Applications

The essence of the cascade effect study is to identify the key factors and clarify the paths between them. The existing literature has proposed various methods for research factors and pathways, both qualitative and quantitative.

The DEMATEL is an effective method for identifying causal relationships in complex systems, involving the assessment of interdependencies between factors [10]. Researchers have used the DEMATEL method in the field of safety risks. For example, Hatefi et al. used the DEMATEL method to assess construction projects and their overall risks [39]. Hui et al. adopted it to analyze the complex interrelationships of enterprise risks [40]. 

The ISM can be used to analyze structural problems in complex situations, transform unclear system models into explicit structural models, and improve system understanding [41]. ISM has been used for factor analyses widely, such as driver analysis [42], vulnerability factor analysis [43], and barrier factor analysis [44]. The ISM considers interactions between factors but neglects to identify critical factors [42].

The SNA method is good at analyzing association relationships among network individuals, which can express the overall characteristics of the network structure and reflect the position of individuals on the whole [9]. Researchers used the SNA method for risk management response to study the risk relationship network of big data projects [45]. Still, it is difficult to grasp the whole picture of the network comprehensively because it over-considers the liaison nature of the network and ignores the isolated points. Therefore, the SNA method plays a complementary and validating role in this paper. That presents another contribution of this study.

In the vast literature on the safety risks of coastal urbanization, limited studies consider the key factors and cascading effects from the intrinsic linkage of risks. Therefore, proposing the model of “Triangular Framework for Safety Risk in New Towns,” this paper analyzes the case of the developing new area of the Lin-gang Special Area of China (Shanghai) Pilot Free Trade Zone (hereinafter referred to as the Lin-gang Special Area). It explores the key factors and cascading effects in the construction and operation process of the new town through the study of disaster-causing factors.

## 3. Methodology

### 3.1. Basis for Integrating DEMATEL-ISM-SNA 

The basis of integrating DEMATEL-ISM-SNA is as follows: 

These methods are all related to matrices and networks. They complement each other so they can be integrated [46,47,48]. First, DEMATEL and SNA effectively identify causal relationships in complex systems. Nevertheless, they cannot clarify the hierarchical structure among factors, while the ISM method remedies this deficiency. Second, the original ISM method is tedious to calculate the reachability matrix K. However, it has previously been observed that the matrix K used in ISM can be obtained by the overall influence matrix H used in DEMATEL [14]. Therefore, integrating DEMATEL and ISM is needed for easy computation [49]. Third, while using the method of DEMATEL and ISM, threshold setting is involved. Experts have developed some threshold-setting strategies but no unified scientific standard [13]. To exclude the influence of the threshold, this study introduced the SNA method. It is good at analyzing the correlations between individual networks and is one of the standard methods for factor analysis. It has previously been observed that degree centrality in the SNA presents the direction of risk transmission on the whole [50]. So we use the SNA method to validate and supplement the hierarchical model [47].

### 3.2. The DEMATEL-ISM-SNA Method

The DEMATEL method obtains the direct relation matrix among factors and calculates the overall influence matrix. Then, the overall influence matrix is converted into the reachability matrix in ISM, and the hierarchical model is divided. Finally, the indicators of SNA are applied to verify and supplement. Steps 1–8 for calculating are listed below. The detailed steps of the DEMATEL-ISM-SNA method can be found in the related literature [51,52,53,54,55]. Figure 2 shows the framework of the DEMATEL-ISM-SNA method.

**Step 1** Identify risk factors, set as, $x_{1}$, $x_{2}$, … …, $x_{n}$. 

**Step 2** Set the influence scale. Determine the influence relationship between factors by the Delphi method, $a_{\mathrm{ij}}$ indicates the direct influence that factor $x_{i}$ has on factor $x_{j}$, using the integer scale below and get the direct relation matrix $M={\left[a_{\mathrm{ij}}\right]}_{n\times n}$.
(1)$$a_{\mathrm{ij}}=\left\{\begin{array}{l} 3,\;\text{very}\;\text{high}\;\text{influence}\;\left(\text{more}\;\text{than}\;\frac{2}{3}\;\text{of}\;\text{experts}\right)\\ 2,\;\text{high}\;\text{influence}\;\left(\frac{1}{3}-\frac{2}{3}\;\text{of}\;\text{experts}\right)\\ 1,\;\text{low}\;\text{influence}\;\left(\text{some}\;\text{but}\;\text{less}\;\text{than}\;\frac{1}{3}\;\text{of}\;\text{experts}\right)\\ 0,\;\text{no}\;\text{influence}\;\left(\text{none}\right) \end{array}\right.$$

**Step 3** The normalized matrix $N={\left[n_{\mathrm{ij}}\right]}_{n\times n}$. This is achieved by using the following equation:(2)$$N=\frac{1}{\underset{1\ll i\ll n}{\max}\sum_{j=1}^{n}a_{\mathrm{ij}}}M$$

**Step 4** The total relation matrix $T={\left[t_{\mathrm{ij}}\right]}_{n\times n}$. It is calculated as (where I is the identity matrix):(3)$$T=N{\left(I-N\right)}^{-1}$$

**Step 5** The overall influence matrix $H={\left[h_{\mathrm{ij}}\right]}_{n\times n}$. This is calculated as:(4)H=T+I

**Step 6** The reachability matrix $K={\left[k_{\mathrm{ij}}\right]}_{n\times n}$. It is computed by using the equation:(5)$$k_{\mathrm{ij}}=\left\{\begin{array}{c} 1,\;h_{\mathrm{ij}}\gg \lambda \\ 0,\;h_{\mathrm{ij}}<\lambda \end{array}\right.$$
where the threshold λ is set by the expert or as appropriate.

**Step 7** Explain the structural modeling. List the reachable set P, and the antecedent set Q of all elements. If $P\left(a_{i}\right)\cap Q\left(a_{i}\right)=P\left(a_{i}\right)$, then this element is classified as the first layer, etc. Finally, the hierarchical structure of the pictorial model is obtained. 

**Step 8** The SNA method is used to verify and further identify key risk factors, which is a validation and essential supplement to the DEMATEL-ISM method. Mainly, centralities are the significant indexes of the SNA approach that measure a node’s importance from many perspectives [56]. Currently, the leading centrality indicators used in the SNA method are degree centrality, betweenness centrality, and proximity centrality. In this paper, degree centrality and betweenness centrality are selected to measure the direct influence relationship and control capability of risk factors. The indicators are calculated as follows: Degree centrality:

Degree centrality measures the number of relationships directly influenced by the node, divided into in-degree and out-degree. 

2.Betweenness centrality.

Betweenness centrality indicates the importance of risks in the network. The larger the centrality value, the stronger the ability to influence other risks and the more critical it is due to more information passing through that factor (assuming that information transfer follows the shortest paths) [55]. When looking for key drivers, we need to focus on aspects with high betweenness centrality [47]. It is calculated by the Formula (6):(6)$$C_{c}\left(i\right)=\underset{j\neq i\neq k\epsilon I}{\sum }\frac{\sigma_{\mathrm{jk}}\left(i\right)}{\sigma_{\mathrm{jk}}}$$
where $\sigma_{\mathrm{jk}}$ is the total number of shortest paths from node j to node k and $\sigma_{\mathrm{jk}}\left(i\right)$ is the number of those paths that pass through the node i.

The following will combine the research on risk management with the actual situation of the Lin-gang Special Area and illustrate the feasibility and effectiveness of the integrated DEMATEL-ISM-SNA method for identifying risk factors and establishing a hierarchical structure model.

## 4. Case Study

### 4.1. Study Area Overview

The Lin-gang Special Area was established in Shanghai on 6 August 2019. Figure 3 shows that it is in southeast Shanghai, adjacent to the East China Sea, with the Dazhi River to its north. It is an important node of the Shanghai Coastal Thoroughfare, with the precise distribution of industrial land, warehousing land, and construction land [57]. 

Through the analysis of typhoons in the East China Sea over the years, existing research recognized that the latitude of the typhoon landfall point varies greatly, with a tendency to move northward [58]. Therefore, the probability of the Lin-gang Special Area being hit by a typhoon will increase. We need to pay attention to this phenomenon. In 2021, under the influence of the typhoon, the Lin-gang Special Area activated emergency plans to transfer susceptible groups and suspend some public transport many times [59,60]. 

The topographic analysis shows that the elevation is basically approximately 4 m or less, and Shanghai’s geological formation primarily comprises soft soil [61]. It has been established that excessive groundwater pumping, the construction of facilities and amenities, and high-rise buildings are the root causes of the land subsidence in Shanghai [62,63,64]. So the control of disasters related to natural and artificial hazards is significant for ensuring the security of the entire urban area.

The planning of the Lin-gang Special Area revolves around two crucial time points, 2025 and 2035 [57]. Industrial development is mainly based on high-tech industries, such as hydrogen energy, biomedicine, and integrated circuit [65]. It will be developed into a high-tech industrial zone and a top research scholar zone, putting forward higher requirements for safety. Hence, it is necessary to grasp the critical risks of the Lin-gang Special Area and develop corresponding measures to guarantee overall security. Against the background of the early stage of construction, a case study in the Lin-gang Special Area was conducted to illustrate the application of the triangular model and systematic analysis method proposed above, aiming to address risk interactions.

### 4.2. The Risk Factors

#### 4.2.1. Identification and Categorization 

The construction and operation of the Lin-gang Special Area is a long process, including many risk factors. There are complex interrelationships between risk factors. Identifying risk factors in the Lin-gang Special Area can help build risk levels. It is important for comprehensive safety planning research. 

Based on the triangle model of safety risk in new towns, combining the existing literature, the policy of the new area, and expert opinions, we identified 16 risk factors.
Typhoon (S1R1): A typhoon is a cyclone with high wind and heavy precipitation, causing casualties and damage [32].Rainstorm and flood (S1R2): Rainstorms and floods often occur in the Yangtze River; intensive rainfall is a key factor in increasing flood hazards [66].Storm surge (S1R3): Storm surge is an increase in sea level caused by low atmospheric pressure and strong winds [67].Land subsidence (S1R4): Land subsidence, vertical land movements, includes natural and human-induced subsidence in sedimentary coastal lowlands [68].Public risk perception (S2R1): Public risk perception is the public’s expressed fear or apprehension about something [69] and is one of the most important factors influencing the government’s management of crisis events [69,70].Perfection of rules and regulations (S2R2): Perfection of rules and regulations refers to the ability to regulate the behavior of people and enterprises [71].Fire (S2R3): Fires include forest, production, and living fires. Considering the industrial structure of the Lin-gang Special Area, the fires covered in this paper refer to those that occurred during the construction and operation [23].Emerging industry risks (S2R4): Introducing new industries such as hydrogen energy and biomedicine in the Lin-gang Special Area will bring different degrees of risk, such as hydrogen leakage and explosion [19,72].Disaster prevention capability of (high-rise) buildings (S3R1): The disaster prevention capability of buildings, especially high-rise buildings, is a protective capability. For example, the new coastal area has high requirements for the corrosion resistance of buildings against the sea wind [73].Population migration (S3R2): Population migration is a future-oriented and predictive issue. Due to population growth, there are risks associated with traffic congestion and relatively weak social services [28,32].Disaster prevention capacity of port facilities (S3R3): Disaster prevention capacity of port facilities refers to the ability to ensure coastal areas avoid coastal erosion and seawater intrusion [73].Disaster prevention capability of infrastructure (S3R4): Disasters, particularly those related to natural hazards, frequently wreak havoc on infrastructure, disrupting transportation, power, and ground communications, and hampering rescue efforts. As urbanization, infrastructure becomes more concentrated, posing additional concerns [72,74].Transportation (S3R5): As an essential area for port transshipment and free trade zone cargo transportation, the Lin-gang Special Area has many safety risks due to collector truck transportation and short barge transportation.Unsafe behaviors of sensitive people (S4R1): People are both disaster-bearing and disaster-causing factors. Unsafe behaviors of sensitive people are the behaviors that affect safety exhibited by this group of people [20].Environmental degradation (S4R2): Environmental degradation is the loss of buffering capacity of local cities from disasters, increasing the risk of exposure [19,23].Infectious diseases (S4R3): Infectious diseases are public health emergencies that affect production and life [75].

#### 4.2.2. Analysis of the Interviewees

Since the assessment of risk factors in the Lin-gang Special Area is related to people’s livelihood and economic stability, requiring the professional knowledge of the assessors, we formed a group of experts and conducted face-to-face interviews. The experts consisted of two college professors, three researchers from Shanghai, and four government officials from the Lin-gang Special Area, all with relevant work experience of more than five years. The interview was divided into three rounds (each expert was interviewed thrice, over 45 min a time). First, we confirmed and supplemented the factors identified in the literature. Second, we summarized experts’ opinions and reached an agreement. Finally, the identified risk factors were judged and scored according to Equation (1). Expert judgment is beneficial to obtaining knowledge of specific aspects of safety risks quickly and accurately.

### 4.3. Results 

The results of performing Step 2 are shown in Appendix A. According to the methods in steps 3 and 4, we obtained the direct relation matrix N and the total relation matrix T. According to the Equations (4) and (5), we calculated the overall impact influence H and the reachability matrix K. 

We found in several experiments that the threshold value of 0.14 was more appropriate. Appendix B shows the final results for the reachability matrix K.

We processed the reachability matrix K according to Step 7. The decomposition table of the first layer is shown in Table 1. 

As shown in Figure 4, the 16 elements are separated into six levels using the ISM approach. We obtained the final stratification results as follows: $L_{1}=\left\{9,11,13\right\};{\;L}_{2}=\left\{12,16\right\};{\;L}_{3}=\left\{2,4,14,15\right\};{\;L}_{4}=\left\{3,7,8,10\right\};{\;L}_{5}=\left\{1,6\right\};{\;L}_{6}=\left\{5\right\}$. 

According to Step 8, the degrees are obtained using UCINET for Windows (version 6), this software is created by Stephen Borgatti, etc. from the University of California, Irvine. Table 2 presents the results of centralities.

### 4.4. Validation

To avoid the impact of the threshold on the result, the obtained hierarchical structure model is validated by the SNA method in this paper. According to the betweenness centrality, the highest figure is environmental degradation (S4R2), which indicates that S4R2 has a high control and a strong ability to influence other factors. Environmental degradation deprives local cities of their buffering capacity of disasters and increases the risk of exposure [23]. A degraded environment also does not provide resources such as clean water and food properly, reducing the ability of people to cope with disasters [76]. The betweenness centrality of typhoon (S1R1) and public risk perception (S2R1) is 0, indicating that the conductivity of these factors in the overall risk network is weak. Therefore, it is necessary to focus on risk factors with high betweenness centrality in risk management, while factors with low betweenness centrality need to be controlled at the source.

From the degree centrality analysis, the risks are usually transferred from the nodes with a higher degree of point-out to those with a higher degree of point-in [50]. The transmission direction of risk factors can be inferred to verify the constructed hierarchical structure model. In this case, the risk transmission relationship of the Lin-gang Special Area is shown in Figure 5, from typhoon (S1R1) and public risk perception (S2R1) to disaster prevention capability of (high-rise) buildings (S3R1), disaster prevention capacity of port facilities (S3R3) and transportation (S3R5), and environmental degradation (S4R2). Furthermore, it should be noted that environmental degradation is not a direct influence in the hierarchical model. Still, it is reasonable because it is the factor with the highest middle centrality and is the most likely to be influenced, as shown in the above analysis. In other words, the results of the centrality analysis are consistent with the constructed hierarchical model, which can prove the model’s reliability.

## 5. Discussion

### 5.1. Hierarchical Structure Analysis

As shown in Figure 4 above, risks can be simplified into three categories: (i) directly influencing factors, (ii) intermediate influencing factors, and (iii) essential influencing factors. The first category is the direct impact factor. It is the most prominent risk factor, represented by the thickened solid line box, including disaster prevention capability of (high-rise) buildings (S3R1), disaster prevention capacity of port facilities (S3R3), and transportation (S3R5). They are buildings, structures, and other entities, and transportation is a factor that cannot be ignored throughout the construction and operation of the new area. 

The third category is the essential factors, which are the intrinsic determinants of the direct influencing factors, represented by the dashed boxes, including typhoon (S1R1), public risk perception (S2R1), and population migration (S3R2). 

The rest belong to the second category of intermediate influencing factors, where the essential influencing factors lead to the intermediate influencing factors. The interaction between the two types of factors eventually leads to direct influencing factors, which bring hazards to the new area.

### 5.2. Suggestions for Safety Risk Control 

The new area transforms a rural area into an emerging city. It is difficult to change entirely in a short period. By clarifying the risk impacts and their relationships, the following recommendations are made:Strengthen the prevention of essential factors. There are three essential factors: typhoon (S1R1), public risk perception (S2R1), and population migration (S3R2). First, according to the previous research, the Lin-gang Special Area may be hit by a typhoon. It will result in high rainfall and flooding secondary disaster chains, jeopardizing infrastructure and transportation. Hence, we must improve our monitoring of dynamic warning and contingency preparations for disasters related to natural hazards [77]. Second, the development is also the process of population importation. We should use the plans to guide the optimal allocation and healthy flow of the population, achieving the harmonious integration of people and the city. Third, we must continue to improve public risk perception and place a high value on human safety education.Implementation of intermediate factors measures. The research shows that most of the factors are intermediate factors. The transmissibility of intermediate factors can influence the upper elements directly or indirectly. Environmental deterioration (S4R2) has the highest betweenness centrality of these variables and should be considered. Studies have found ecological regulation to help lower risks [78,79].Pay attention to direct factor control. There are three essential factors: disaster prevention capability of (high-rise) buildings (S3R1), disaster prevention capacity of port facilities (S3R3), and transportation (S3R5). The Lin-gang Special Area is coastal and has soft ground, so it is not advisable to build many high-rise buildings. It should be planned reasonably to improve the existing high-rise buildings’ disaster prevention standards and capability. To make coastal high-rise buildings typhoon-resistant, it is important to pay attention to their gaps. It is recommended to avoid using rust-prone materials to prevent the influence of sea breeze and tide on buildings and structures. When building ports in the Lin-gang Special Area, attention shall be paid to the safety protection of the sea pond infrastructure. For example, implement the control of the sea ponds, establish a scientific index system, build a comprehensive safety plan and emergency plan to integrate sea and sky, and improve the port facilities’ disaster prevention capacity. Ultimately, the emphasis on transportation should be maintained throughout the city’s development and administration.

### 5.3. Innovation

This study uncovers various safety risks and their interactions in the Lin-gang Special Area and provides an innovative tool for managing them. The case study in the Lin-gang Special Area demonstrates that the integrated strategy described in this research is a valuable tool for dealing with both traditional and developing safety hazards. It does an excellent job of finding critical factors to be mitigated with priority. 

The main innovations of this paper are as follows:This study defines the concept of safety risk in new coastal towns and proposes a triangle model to identify and assess risk relationships. This model fully considers the risks exposed in the process of urban construction and industrial development.This research addresses the constraints of previous approaches for computing and assessing risk interactions, excluding the influence of artificially set thresholds on the results. The SNA plays a complementary and validating role in this paper

## 6. Conclusions

This study aims to improve new towns’ disaster prevention capacity and ensure sustainable development. To lessen the adverse effects in the area, we propose a triangle model of safety risks in the new towns and an integrated DEMATEL-ISM-SNA method. A case study in an urbanized city whose function and geographical location are critical in China is conducted to present its practical application. This strategy can classify risk variables and intercept them before they occur by quickly converting expert knowledge into useful information.

Taking the Lin-gang Special Area as a case study, we clarified 16 risk factors and their hierarchical structure. Results show that:The most potent driving forces and essential elements are typhoon, storm surge, public risk perception, and population migration.Three factors, disaster prevention capability of (high-rise) buildings, disaster prevention capacity of port facilities, and transportation are at the top of the ISM model. They have the most direct influence on the new town.Environmental degradation has the highest betweenness centrality among all factors and is the most transmissible.

Therefore, we should pay special attention to these factors during the construction and operation of new coastal towns. The approach presented in this study can help find these key factors while constructing new coastal towns. It is equipped with theoretical and practical significance. First, the findings are helpful for researchers in the field of new coastal towns because they provide a list of crucial factors and the connections among them that aid in implementing the construction plan with a safety-focused perspective. Second, the findings can assist managers and policymakers in the Lin-gang Special Area by speeding up decision making in identifying crucial elements for development. The results of this integrated approach guide decision-makers to focus on the most critical elements and select the best plan based on the hierarchical structure of risk factors. Therefore, this approach could be extended to other similar coastal new towns in the future. 

This study has some limitations. First, the sample in this paper is mainly obtained from expert opinions. In the future, it can be extended to incorporate multi-case analysis. Second, the current research, such as proposing a safety risk framework and an integrated method, is the initial step for urbanization safety risk governance. For further study, we may study in a deeper field, e.g., at the technical or management level, its safety risks can be identified and controlled.

## Figures and Tables

**Figure 1 ijerph-19-10496-f001:**
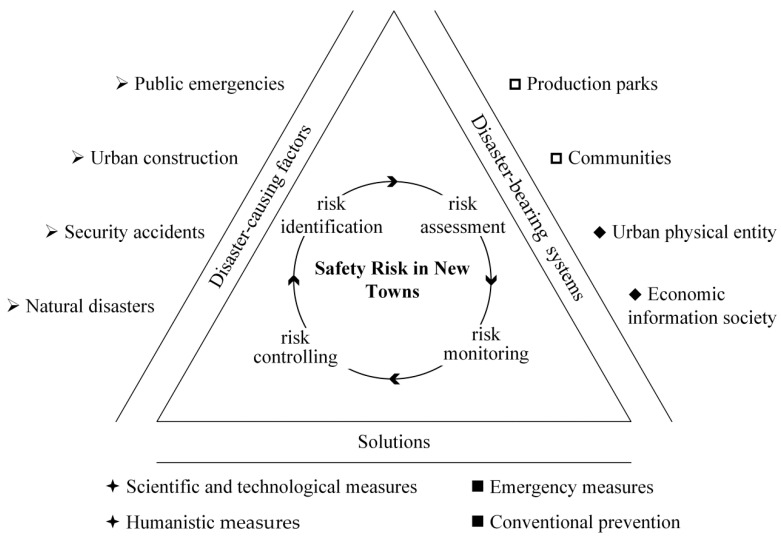
The triangular framework for safety risk in new towns.

**Figure 2 ijerph-19-10496-f002:**
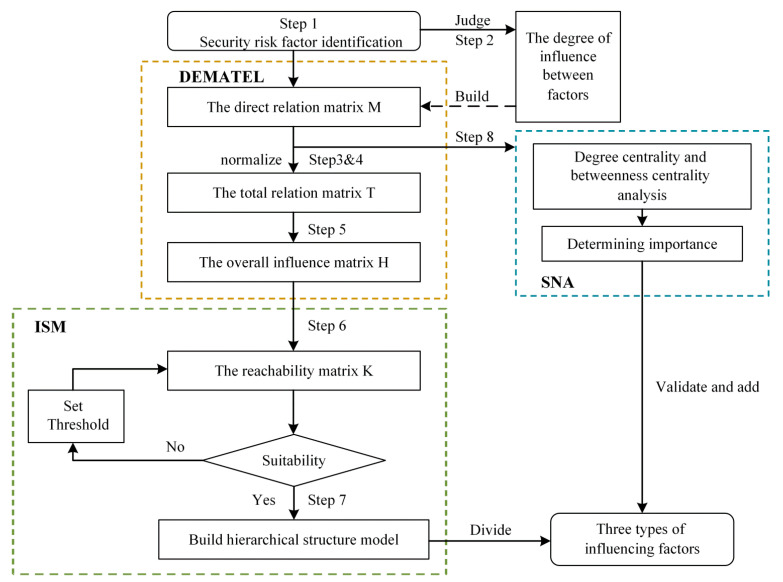
The framework of the DEMATEL-ISM-SNA method.

**Figure 3 ijerph-19-10496-f003:**
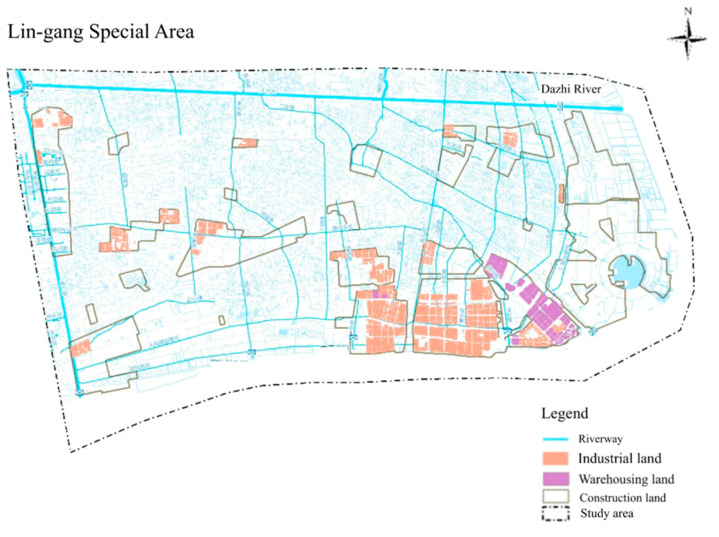
Study area.

**Figure 4 ijerph-19-10496-f004:**
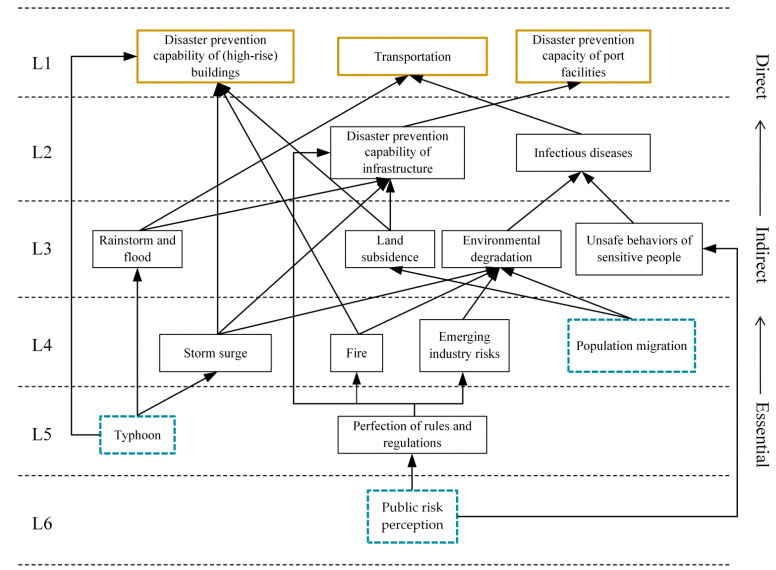
The hierarchical structure of the risk factors in the Lin-gang Special Area.

**Figure 5 ijerph-19-10496-f005:**
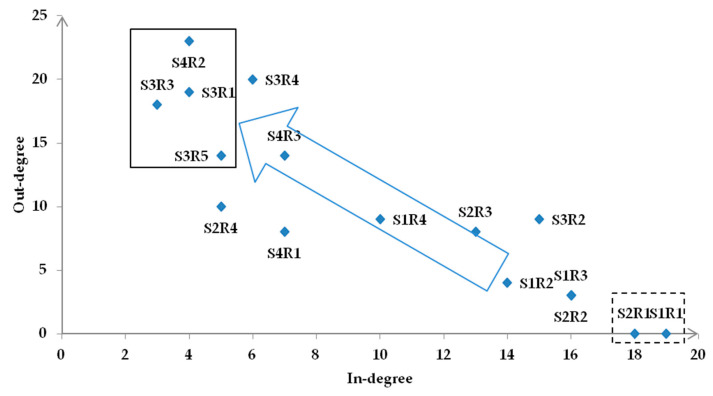
Transmission map of the relationship among safety risk factors in the Lin-gang Special Area.

**Table 1 ijerph-19-10496-t001:** First layer decomposition.

Factors	Reachable Set P(a_i_)	Antecedent Set Q(a_i_)	P(a_i_)∩Q(a_i_)
$a_{1}$	1, 2, 3, 9, 11, 12, 13, 15	1	1
$a_{2}$	2, 11, 12, 13	1, 2	2
$a_{3}$	3, 9, 11, 12, 15	1, 3	3
$a_{4}$	4, 9, 11, 12	4, 10	4
$a_{5}$	5, 6, 7, 8, 13, 14, 15, 16	5	5
$a_{6}$	6, 7, 8, 12, 15	5, 6	6
$a_{7}$	7, 9, 15	5, 6, 7	7
$a_{8}$	8, 15	5, 6, 8	8
$a_{9}$	9	1, 3, 4, 7, 9, 10	9
$a_{10}$	4, 9, 10, 12, 15, 16	10	10
$a_{11}$	11	1, 2, 3, 4, 11, 12	11
$a_{12}$	11, 12	1, 2, 3, 4, 6, 10, 12	12
$a_{13}$	13	1, 2, 5, 13, 16	13
$a_{14}$	14, 16	5, 14	14
$a_{15}$	15, 16	1, 3, 5, 6, 7, 8, 10, 15	15
$a_{16}$	13, 16	5, 10, 14, 15, 16	16

**Table 2 ijerph-19-10496-t002:** Crisp values of out (in)-degree and betweenness centrality.

	Out-Degree	In-Degree	Betweenness Centrality
S1R1	19	0	0
S1R2	14	4	9.917
S1R3	16	3	0.25
S1R4	10	9	2.45
S2R1	18	0	0
S2R2	16	3	0.533
S2R3	13	8	9.783
S2R4	5	10	6.85
S3R1	4	19	11.95
S3R2	15	9	17.1
S3R3	3	18	4.2
S3R4	6	20	7.25
S3R5	5	14	6.517
S4R1	7	8	3.483
S4R2	4	23	25.183
S4R3	7	14	14.533

## Data Availability

Data sharing is not applicable.

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
