# Peer review of "Hierarchical Structure Model of Safety Risk Factors in New Coastal Towns: A Systematic Analysis Using the DEMATEL-ISM-SNA Method"

_ijerph, 2022, doi:10.3390/ijerph191710496_

Round 1

Reviewer 1 Report

Thank you for the opportunity to review the paper. It is an interesting piece of work. In my opinion it needs just some polishing before being published.

Line: 42: you cannot write „new region”. This term states rather for geographical area. I assume you had in mind newly built/urbanized areas. Region is much broader. Please correct it.

Lines 46-50: I am quite confused by this statement. As I understood these new coastal towns are constructed based on planning. Therefore my question is why this will lead to overlapping old and new threats? Ok, I acknowledge the fact that there is rising complexity of those local systems, there is increasing population density and so on. But by proper planning you may exclude some threats. Maybe you could elaborate a bit more on that to dispel any doubts.

Lines 53-54: You may add here resilience arrangements analysis or policy arrangements approach (for example 10.3390/su14042052) as well.

Fig. 1. Pleas explain use of different bullets. Also move solution bullets so the are not almost upside down. Now it is hard to read.

Lines 215-219: It is really necessary to give here a bit wider climatologic background. How many and of what scale natural disasters hit that spot in recent decades?

Fig. 2. “Jude” at Step 2, please correct it

4.2.2 – How many interviews you have conducted (have you interviewed only the mentioned experts)? How long did an interview take? Please elaborate more on that element of your study.

Author Response

Manuscript Number: ijerph-1825009

Title: Hierarchical Structure Model of Safety Risk Factors in New Coastal Towns: A Systematic Analysis Using DEMATEL-ISM-SNA Method

Dear Editors and Reviewers,

Thank you for giving us a chance to revise our manuscript entitled "Hierarchical Structure Model of Safety Risk Factors in New Coastal Towns: A Systematic Analysis Using DEMATEL-ISM-SNA Method" (Manuscript ID: ijerph-1825009). And also, thank the reviewers for the valuable comments, which gives us excellent help in improving our manuscript.  

According to the reviewers’ comments, we have made major revisions and addressed the reviewers’ comments point by point in the response. To provide a better reading experience, the response is in orange, the revised text abstracted from the manuscript is in blue, and its page and line locations are highlighted in yellow to facilitate the review process.

Thank you again for your advice, attention, and timely response. We are looking forward to hearing from you at your earliest convenience.

Best regards,

The authors

Response to Reviewer 1 Comments:

Point 1: Line 42: you cannot write “new region”. This term states rather for geographical area. I assume you had in mind newly built/urbanized areas. Region is much broader. Please correct it.

[Response]: Thank you for your comments. We agree with your suggestion that the definition of the region is much broader. So we have replaced the term “region” with “area” in our manuscript (Page 1, Line 42). Simultaneously, the term “region” has also been adjusted accordingly (Page 13, Line 449).

Point 2: Lines 46-50: I am quite confused by this statement. As I understood these new coastal towns are constructed based on planning. Therefore my question is why this will lead to overlapping old and new threats? Ok, I acknowledge the fact that there is rising complexity of those local systems, there is increasing population density and so on. But by proper planning you may exclude some threats. Maybe you could elaborate a bit more on that to dispel any doubts.

[Response]: Thank you very much for pointing this out. This suggestion was very helpful in improving our manuscript. Planning, as you said, is meticulous and comprehensive, and indeed there are special sections on disaster prevention design. But now, many plans are not systematic enough, so they are not better at avoiding increased risk. Of course, there are different focuses of planning, and our team is also doing comprehensive disaster mitigation planning, which focuses on risk reduction. Therefore, identifying the risk coupling in building new coastal towns and breaking this coupling effect is the meaning for our team to do so. We hope to provide effective suggestions for the Lin-gang Special Area and other new town construction. We have made a detailed addition in Section 1. The new sentence is shown below.

  • Introduction (Page 2, Line 50-56): Urban planning is an effective measure to provide structure to activities through long-term thinking and decision-making to guide future action [1]. It does have a special section on disaster prevention and mitigation. But now, many plans are not systematic and targeted enough to avoid risks [2]. Therefore, we carry out comprehensive disaster prevention and mitigation planning research. The aim is to identify the risk coupling in building new coastal towns and break this coupling effect. We hope to provide practical suggestions for new coastal town construction.

References

  1. Bush, J.; Doyon, A. Building urban resilience with nature-based solutions: How can urban planning contribute? Cities 2019, 95, 102483.102481-102483.102488.
  2. Nguyen, T.T.; Ngo, H.H.; Guo, W.; Wang, X.C.; Ren, N.; Li, G.; Ding, J.; Liang, H. Implementation of a specific urban water management - Sponge City. Science of The Total Environment 2019, 652, 147-162.

Point 3. Lines 53-54: You may add here resilience arrangements analysis or policy arrangements approach (for example 10.3390/su14042052) as well.

[Response]: Thank you for your comments. In Section 1 of the new manuscript, we have added the other approach, such as resilience arrangements analysis (Page 2, Line 58-59).

Point 4: Fig. 1. Pleas explain use of different bullets. Also move solution bullets so they are not almost upside down. Now it is hard to read.

[Response]: Thank you for your comments. We have rewritten this section and adjusted the position of the text (Page 3, Line 120), making it easy to read. The new sentence is shown below.

  • 2.2 Triangular Framework for Pubilc Security and Technology (Page 3, Line 116-119): As indicated in Figure 1, the triangle public safety science and technology model is applied to the safety risk of new coastal towns, including disaster-causing factors, disas-ter-bearing systems, and solutions. Only by coordinating these parts, formulating plans, and implementing them can comprehensively reduce safety risks [1].

Reference

  1. Weicheng, F.A.N.; Yi, L.I.U.; Wenguo, W. Triangular Framework and "4+1" Methodology for Public Security Science and Technology. Science & Technology Review 2009, 27, 3-3.

Point 5: Lines 215-219: It is really necessary to give here a bit wider climatologic background. How many and of what scale natural disasters hit that spot in recent decades?

[Response]: Thank you very much for pointing this out. This suggestion was very helpful for improving our manuscript. We are sorry for the ambiguity in the previous expression. We have rewritten this section. The new sentence is shown below.

  • 4.1 Study Area Overview (Page 7, Line 247-259): Through the analysis of typhoons in the East China Sea over the years, existing research recognized that the latitude of the typhoon landfall point varies greatly, with a tendency to move northward [1]. Therefore, the probability of the Lin-gang Special Area being hit by a typhoon will increase. We need to pay attention to this phenomenon. In 2021, under the influence of the typhoon, the Lin-gang Special Area activated emergency plans to transfer susceptible groups and suspend some public transport many times [2,3].

The topographic analysis shows that the elevation is basically around 4m or less, and Shanghai's geological formation primarily comprises soft soil [4]. It has been established that excessive groundwater pumping, the construction of facilities and amenities, and high-rise buildings are the root causes of the land subsidence in Shanghai [5-7]. So the control of disasters related to natural and artificial hazards is significant for ensuring the security of the entire urban area.

References

  1. Lu, X.; Dong, C.; Li, G. Variations of typhoon frequency and landfall position in East China Sea from 1951 to 2015. Transactions of Atmospheric Sciences 2018, 41, 433-440.
  2. Available online: http://sh.cma.gov.cn/sh/news/qxyw/202109/t20210913_3813856.html (accessed on 3 August).
  3. Available online: http://sh.cma.gov.cn/sh/news/qxyw/202107/t20210724_3562269.html (accessed on 3 August).
  4. Lyu, H.M.; Shen, S.L.; Zhou, A.; Yang, J. Risk assessment of mega-city infrastructures related to land subsidence using improved trapezoidal FAHP. Science of The Total Environment 2019, 717, 135310.
  5. Ye-Shuang, X.; Shui-Long, S.; Dong-Jie, R.; Huai-Na, W. Analysis of Factors in Land Subsidence in Shanghai: A View Based on a Strategic Environmental Assessment. Sustainability 2016, 8, 573.
  6. Wu, H.-N.; Shen, S.-L.; Yang, J. Identification of Tunnel Settlement Caused by Land Subsidence in Soft Deposit of Shanghai. Journal of Performance of Constructed Facilities 2017, 31, doi:10.1061/(asce)cf.1943-5509.0001082.
  7. Zhao, Y.; Zhou, L.; Wang, C.; Li, J.; Qin, J.; Sheng, H.; Huang, L.; Li, X. Analysis of the Spatial and Temporal Evolution of Land Subsidence in Wuhan, China from 2017 to 2021. Remote Sensing 2022, 14, doi:10.3390/rs14133142.

Point 6: Fig. 2. “Jude” at Step 2, please correct it

[Response]: Thank you very much for pointing this out. This suggestion was very helpful for improving our manuscript. We are sorry for our inaccurate language expressions in the previous version. We have replaced the word “Jude” with “Judge” in Figure 2 (Page 5, Line 199).

Point 7: 4.2.2 – How many interviews you have conducted (have you interviewed only the mentioned experts)? How long did an interview take? Please elaborate more on that element of your study.

[Response]: Thank you so much for your valuable suggestions. We have interviewed nine experts who have experience in urban safety. They identified and rated risk factors in the interview. The interview was divided into three rounds (each expert was interviewed three times, over 45 minutes a time). First, we confirmed and supplemented the factors identified in the literature. Second, we summarized experts' opinions and reached an agreement. Finally, the identified risk factors were judged and scored according to equation (1). We have revised the sentence. The new sentence is shown below.

  • 4.2.2 Analysis of the interviewees (Page 9, Line 323-333): Since the assessment of risk factors in the Lin-gang Special Area is related to people's livelihood and economic stability, requiring the professional knowledge of the assessors, we formed a group of experts and conducted face-to-face interviews. The experts consisted of two college professors, three researchers from Shanghai, and four government officials from the Lin-gang Special Area, all with relevant work experience of more than five years. The interview was divided into three rounds (each expert was interviewed three times, over 45 minutes a time). First, we confirmed and supplemented the factors identified in the literature. Second, we summarized experts' opinions and reached an agreement. Finally, the identified risk factors were judged and scored according to equation (1). Expert judgment is beneficial to obtaining knowledge of specific aspects of safety risks quickly and accurately.

Reviewer 2 Report

Dear authors,

Please consider the following in revising your manuscript:

1. Elaborate need for the integration of the three methods.

2. What softwares were used for the calculations?

3. The THK matrix or the similarities of the outputs of DEMATEL, ISM, and SNA do not automatically justify the practical need for using this integrated method. Please provide further discussion.

4. What aspects of each method support or contradict each other? 

5. Are Tables 1 and 2 essential in the manuscript? Provide their interpretation, relevance and implications to this paper.

6. Elaborate your interpretation of Table 4. Provide literature when to consider intermediate centrality high or low.

7. Based on the results, were you able to demonstrate all mentioned advantage/s of the integrated methods for this study? Please elaborate.

Author Response

Manuscript Number: ijerph-1825009

Title: Hierarchical Structure Model of Safety Risk Factors in New Coastal Towns: A Systematic Analysis Using DEMATEL-ISM-SNA Method

Dear Editors and Reviewers,

Thank you for giving us a chance to revise our manuscript entitled "Hierarchical Structure Model of Safety Risk Factors in New Coastal Towns: A Systematic Analysis Using DEMATEL-ISM-SNA Method" (Manuscript ID: ijerph-1825009). And also, thank the reviewers for the valuable comments, which gives us excellent help in improving our manuscript.  

According to the reviewers’ comments, we have made major revisions and addressed the reviewers’ comments point by point in the response. To provide a better reading experience, the response is in orange, the revised text abstracted from the manuscript is in blue, and its page and line locations are highlighted in yellow to facilitate the review process.

Thank you again for your advice, attention, and timely response. We are looking forward to hearing from you at your earliest convenience.

Best regards,

The authors

Response to Reviewer 2 Comments:

Point 1: Elaborate need for the integration of the three methods.

[Response]: Thank you for your comments. We have three reasons for integrating three methods.

First, Decision Making Trial and Evaluation Laboratory (DEMATEL) and Social Network Analysis (SNA) are effective methods for identifying causal relationships in complex systems. But they cannot clarify the hierarchical structure among factors, while Interpretive Structural Modeling (ISM) remedies this deficiency.

Second, the original ISM method is tedious to calculate the reachable matrix. However, the reachable matrix used in ISM analysis can be obtained by DEMATEL. Therefore, integrating DEMATEL and ISM is needed for easy computation.

Third, while using the method of DEMATEL and ISM, threshold setting is involved. Although experts have developed some threshold-setting strategies, no unified scientific standard exists. To exclude the influence of the threshold, this study introduced the SNA method. It is good at analyzing the correlations between individual networks and is one of the common methods for factor analysis. It has previously been observed that degree centrality in the SNA presents the direction of risk transmission. So we use the SNA method to validate and supplement the hierarchical model. Above, the idea and the need to integrate the three methods were described, and we have rewritten section 3.1 (Page 5, Line 177-191).

Point 2: What softwares were used for the calculations?

[Response]: Thank you for your comments. In the DEMATEL-ISM-SNA method, we have designed eight steps. Among them, Steps 5-8 involve calculation. Excel was used in steps 5-7, and UCINET for Windows (version 6) was used in step 8.

Point 3: The THK matrix or the similarities of the outputs of DEMATEL, ISM, and SNA do not automatically justify the practical need for using this integrated method. Please provide further discussion.

[Response]: Thank you very much for pointing this out, this suggestion was very helpful in improving our manuscript. We have rewritten section 3.1. The new sentence is shown below.

  • Section 3.1 (Page 5, Line 177-191): These methods are all related to matrices and networks. They complement each other so they can be integrated [1-3]. First, DEMATEL and SNA effectively identify causal relationships in complex systems. But they cannot clarify the hierarchical structure among factors, while the ISM method exactly remedies this deficiency. Second, the original ISM method is tedious to calculate the reachability matrix K. However, it has previously been observed that the matrix K used in ISM can be obtained by the overall influence matrix H used in DEMATEL [4]. Therefore, integrating DEMATEL and ISM is needed for easy computation [5]. Third, while using the method of DEMATEL and ISM, threshold setting is involved. Experts have developed some threshold-setting strategies but no unified scientific standard [6]. To exclude the influence of the threshold, this study introduced the SNA method. It is good at analyzing the correlations between individual networks and is one of the standard methods for factor analysis. It has previously been observed that degree centrality in the SNA presents the direction of risk transmission on the whole [7]. So we use the SNA method to validate and supplement the hierarchical model [2].

References

  1. Kamble, S.S.; Gunasekaran, A.; Sharma, R. Modeling the blockchain enabled traceability in agriculture supply chain. International Journal of Information Management 2020, 52, doi:10.1016/j.ijinfomgt.2019.05.023.
  2. Ko, S.-S.; Ko, N.; Kim, D.; Park, H.; Yoon, J. Analyzing technology impact networks for R&D planning using patents: combined application of network approaches. Scientometrics 2014, 101, 917-936, doi:10.1007/s11192-014-1343-2.
  3. Pandey, P.; Agrawal, N.; Saharan, T.; Raut, R.D. Impact of human resource management practices on TQM: an ISM-DEMATEL approach. Tqm Journal 2022, 34, 199-228, doi:10.1108/tqm-03-2021-0095.
  4. Zhou, D.; Zhang, L. Establishing hierarchy structure in complex systems based on the integration of DEMATEL and ISM. Journal of Management Sciences in China 2008, 11, 20-26.
  5. Zhou, D.Q.; Zhang, L.; Li, H.W. A Study of the System's Hierarchical Structure Through Integration of Dematel and ISM. In Proceedings of the International Conference on Machine Learning & Cybernetics, 2006.
  6. Chen, J.K. Improved DEMATEL-ISM integration approach for complex systems. PLOS ONE 2021, 16.
  7. Yang, R.J.; Zou, P.X.W. Stakeholder-associated risks and their interactions in complex green building projects: A social network model. Building and Environment 2014, 73, 208-222, doi:10.1016/j.buildenv.2013.12.014.

Point 4: What aspects of each method support or contradict each other?

[Response]: Thank you for your valuable comments. The reasons are in the following.

First, the overall influence matrix H obtained by the DEMATEL method and the reachable matrix K obtained by the ISM method have common points: the non-zero elements of both reflect the existence of mutual influence relationships among factors, and the zero elements reflect the absence of influence relationships among the corresponding factors. The difference is that matrix H contains more information than matrix K. The matrix H reflects not only the existence of the influence relationship between factors but also the degree of influence between factors. But matrix K only demonstrates the existence of the influence relationship. Therefore, there is a single mapping between the matrix H and the matrix K: the matrix H calculated by the DEMATEL method can be used to obtain the matrix K [1].

Second, the hierarchical structure model obtained by the ISM method and the point degree (centrality) acquired by the SNA method have a commonality: the hierarchical structure model reveals the relationship between factors. And SNA is good at analyzing the correlations between individual networks. It has previously been observed that degree centrality in the SNA presents the direction of risk transmission [2]. So we can use the SNA method to exclude the influence of the threshold, validating the effectiveness of the ISM. We have revised the sentence. The new sentence is shown below.

  • Section 3.1 (Page 5, Line 180-183): Second, the original ISM method is tedious to calculate the reachability matrix K. However, it has previously been observed that the matrix K used in ISM can be obtained by the overall influence matrix H used in DEMATEL [1].
  • Section 3.1 (Page 5, Line 188-180): It has previously been observed that degree centrality in the SNA presents the direction of risk transmission on the whole [2]. So we use the SNA method to validate and supplement the hierarchical model.

References

  1. Zhou, D.; Zhang, L. Establishing hierarchy structure in complex systems based on the integration of DEMATEL and ISM. Journal of Management Sciences in China 2008, 11, 20-26.
  2. Yang, R.J.; Zou, P.X.W. Stakeholder-associated risks and their interactions in complex green building projects: A social network model. Building and Environment 2014, 73, 208-222, doi:10.1016/j.buildenv.2013.12.014.

Point 5: Are Tables 1 and 2 essential in the manuscript? Provide their interpretation, relevance and implications to this paper.

[Response]: Thank you for your valuable comments. We have put Tables 1 and 2 in the appendix (Page 14-15, Line 495-499) to show the calculation process clearer to readers. Table 1 shows the results of the original data processing. It is the beginning of the calculations using the DEMATEL-ISM-SNA method. Table 2 shows the results of the reachable matrix. The hierarchical model was constructed based on Table 2.

Point 6: Elaborate your interpretation of Table 4. Provide literature when to consider intermediate centrality high or low.

[Response]: Thank you so much for your valuable suggestions. Table 4 presents the results of centralities mentioned in Section 3.2 (step 8). Now, it is named table 2 because the original table 1 and 2 have been put in the appendix for brevity. We have revised section 3.2 to show when to consider intermediate centrality high or low.

First, centralities are the significant indexes of the SNA approach that measure a node's importance from many perspectives [1]. Besides, to be consistent with the expression in the existing literature [1,2], we replaced "intermediate centrality" with "betweenness centrality" throughout the manuscript.

Second, a factor with higher betweenness centrality has significant control and influence on the network due to more information passing through that factor (assuming that information transfer follows the shortest paths). When looking for key drivers, we need to focus on aspects with high betweenness centrality [2,3]. The new sentence is shown below.

  • Section 3.2 (Page 6, Line 315-319): Betweenness centrality indicates the importance of risks in the network. The larger the centrality value, the stronger the ability to influence other risks and the more critical it is due to more information passing through that factor (assuming that information transfer follows the shortest paths) [2]. When looking for key drivers, we need to focus on aspects with high betweenness centrality [3]. It is calculated by the formula (6).

References

  1. Kim, E.; Cho, Y.; Kim, W. Dynamic patterns of technological convergence in printed electronics technologies: patent citation network. Scientometrics 2014, 98, 975-998.
  2. Dehdasht, G.; Ferwati, M.S.; Mohandes, S.R.; El-Sabek, L.; Edwards, D.J. Towards expediting the implementation of sustainable and successful lean paradigm for construction projects: a hybrid of DEMATEL and SNA approach. Engineering Construction and Architectural Management 2022, doi:10.1108/ecam-02-2022-0121.
  3. Ko, S.-S.; Ko, N.; Kim, D.; Park, H.; Yoon, J. Analyzing technology impact networks for R&D planning using patents: combined application of network approaches. Scientometrics 2014, 101, 917-936, doi:10.1007/s11192-014-1343-2.

Point 7: Based on the results, were you able to demonstrate all mentioned advantages of the integrated methods for this study? Please elaborate.

[Response]: Thank you for your valuable comments. We have revised Section 6 (Page 14, Line 467-476). First, the results of this study show that the integrated methods can simplify the calculation. As long as we use this method, we can obtain the hierarchical structure of the risk factors in eight steps. Second, this strategy is effective in safety risk analysis. It can classify risk variables and intercept them before they occur by converting expert knowledge into useful information. Third, the results demonstrated in Figure 4 (Page 10, Line 350) and Table 2 (Page 11, Line 353) are beneficial for researchers in the field of new coastal towns because they present a list of critical factors and the association among crucial factors which helps its construction plan implementation with a safety-oriented focus.

Moreover, the results can help managers and policymakers in the Lin-gang Special Area by boosting the decision-making process in identifying critical factors for implementation. The outcome of the integrated methods guides decision-makers to concentrate on the most significant aspects and select the optimum strategy based on the hierarchical structure of the risk factors. Therefore, it has theoretical and practical significance to approach the safety risk of new towns by using this integrated method. The new sentence is shown below.

  • Section 6 (Page 14, Line 467-476): It is equipped with theoretical and practical significance. First, the findings are helpful for researchers in the field of new coastal towns because they provide a list of crucial factors and the connections among them that aid in the implementation of the construction plan with a safety-focused perspective. Second, the findings can assist managers and policymakers in the Lin-gang Special Area by speeding up decision-making in identifying crucial elements for development. The results of this integrated approach guide decision-makers to focus on the most critical elements and select the best plan based on the hierarchical structure of risk factors. Therefore, this approach could be extended to other similar coastal new towns in the future.

Reviewer 3 Report

Dear Authors,

Please find my comments in the attached pdf file

With best regards

Author Response

Manuscript Number: ijerph-1825009

Title: Hierarchical Structure Model of Safety Risk Factors in New Coastal Towns: A Systematic Analysis Using DEMATEL-ISM-SNA Method

Dear Editors and Reviewers,

Thank you for giving us a chance to revise our manuscript entitled "Hierarchical Structure Model of Safety Risk Factors in New Coastal Towns: A Systematic Analysis Using DEMATEL-ISM-SNA Method" (Manuscript ID: ijerph-1825009). And also, thank the reviewers for the valuable comments, which gives us excellent help in improving our manuscript.  

According to the reviewers’ comments, we have made major revisions and addressed the reviewers’ comments point by point in the response. To provide a better reading experience, the response is in orange, the revised text abstracted from the manuscript is in blue, and its page and line locations are highlighted in yellow to facilitate the review process.

Thank you again for your advice, attention, and timely response. We are looking forward to hearing from you at your earliest convenience.

Best regards,

The authors

Response to Reviewer 3 Comments:

We have this response structure to your detailed review:

Point x: Page x, Line x (in the original manuscript), reviewer’s comments.

[Response]: xxx (Page x and Line x mean the location in the new manuscript)

Point 1: Page 1, Line 39, I suggest to replace "natural disasters" with "disasters related to natural hazards". I suggest to apply this replacement throughout the manuscript where needed.

[Response]: Thank you so much for your valuable suggestions. We have replaced “natural disasters” with “disasters related to natural hazards” in our manuscript (Page 1, Line 39). Simultaneously, the “natural disasters” have been adjusted accordingly (Page 2, Line 45; Page 12, Line 405). Also, “Disasters, particularly natural disasters” has been replaced “Disasters, particularly those related to natural hazards” (Page 8, Line 308).

Point 2: Page 1, Line 42, I suggest to replace "catastrophes" with "disasters". I suggest to apply this replacement throughout the manuscript where needed.

[Response]: Thank you so much for your valuable suggestions. We have replaced “catastrophe” with “disaster” in our manuscript (Page 1, Line 43). Simultaneously, the “catastrophe” has been adjusted accordingly (Page 12, Line 607).

Point 3: Page 2, Line 45, please see previous comments.

[Response]: Thank you so much for your valuable suggestions. We have replaced “natural disasters” with “disasters related to natural hazards” in our manuscript (Page 1, Line 39). Simultaneously, the “natural disasters” have been adjusted accordingly (Page 2, Line 45; Page 12, Line 405). Also, “Disasters, particularly natural disasters” has been replaced “Disasters, particularly those related to natural hazards” (Page 8, Line 308).

Point 4: Page 2, Line 51-52, I suggest to cite the studies develop specific methods and tools to capture safety.

[Response]: Thank you for your valuable comments. In Section 1 of the new manuscript, we have added the other approach, such as resilience arrangements analysis (Page 2, Line 58-59).

Point 5: Page 2, Line 52-63, there is a lack of citations in this paragraph. I suggest to cite relevant studies throughout the paragraph.

[Response]: Thank you for your valuable comments. We have revised this section, citing relevant studies in this paragraph (Page 2, Line 59-65).

Point 6: Page 2, Line 85-86, please add a space before parenthesis.

[Response]: We have added the space before parenthesis in the paper (Page 2, Line 93-94).

Point 7: Page 3, Line 103, I suggest to use lowercase.

[Response]: Thank you for your detailed review. We have used lowercase in the paper (Page 3, Line 111).

Point 8: Figure 1, the image is of low analysis. Please replace with a figure with higher analysis, make the letters bigger and reverse the words in the lower part of the figure. The reader should turn the page in order to read the figure.

[Response]: Thank you very much for pointing this out, this suggestion was very helpful for improving our manuscript. We have replaced it with a figure with higher analysis and adjusted the position of the text (Page 3, Line 120).

Point 9: Page 3, Line 113, please define S1, S2, S3 and S4. I did not find their explanation in the text.

[Response]: Thank you for your comments. We have defined S1, S2, S3 and S4. I did not find their explanation in the text. The new sentence is shown below.

  • 2.2 Triangular Framework for Pubilc Security and Technology (Page 3, Line 123-132): This study summarizes four categories for security in new towns based on previous research [1-4].
  1. Natural disasters (S1): Natural disasters refer to disasters related to natural hazards which occur or are likely to happen in this area and cause heavy losses.
  2. Safety accidents (S2): Safety incidents caused by the industry or bring risks to the industry.
  3. Urban construction (S3): Urban construction comes with long-term site safety risks, population migration, risks posed by buildings, and other related risks.
  4. Public events (S4): Public events refer to events widely concerned or followed up by social groups at different levels.

References

  1. Weicheng, F.A.N.; Yi, L.I.U.; Wenguo, W. Triangular Framework and "4+1" Methodology for Public Security Science and Technology. Science & Technology Review 2009, 27, 3-3.
  2. Zhong, S.; Fang, Z.; Zhu, M.; Huang, Q. A geo-ontology-based approach to decision-making in emergency management of meteorological disasters. Natural Hazards 2017, 89, 531-554, doi:10.1007/s11069-017-2979-z.
  3. Wang, Z.; Kong, W.; Fang, D.; Duan, Z. Research on urban flood and waterlog emergency scenario deduction based on Bayesian network. China Safety Science Journal(CSSJ) 2021, 31, 182-188.
  4. Zheng, Y.; Capra, L.; Wolfson, O.; Yang, H. Urban Computing: Concepts, Methodologies, and Applications. Acm Transactions on Intelligent Systems and Technology 2014, 5, doi:10.1145/2629592.

Point 10: Figure 2, the image is of low analysis. Please replace with a figure with higher analysis, and make the letters bigger. Please use different colors in different steps of the flowchart.

[Response]: Thank you for your comments. We have replaced it with a figure with higher analysis (Page 5, Line 199).

Point 11: Page 6, Line 211, I suggest to replace with "km2".

[Response]: Thank you for your comments. We have replaced “square kilometers” with “km2” in our manuscript (Page 7, Line 242).

Point 12: Figure 3, I suggest to add a map of the study area with grid in order to show the coordinates of the area and a north arrow for the orientation.

[Response]: Thank you very much for pointing this out. This suggestion was very helpful for improving our manuscript. We have revised Section 4.1 and remade a map with the grid to show the coordinates of the area and a north arrow for the orientation. In addition, we have demonstrated some functional divisions in the map to clarify the discussion. The new sentence is shown below.

  • 4.1 Study Area Overview (Page 7, Line 240-244): Figure 3 shows that the special area is in southeast Shanghai, adjacent to the East China Sea, with the Dazhi River to its north and a total area of 873 km2. It is an important node of the Shanghai Coastal Thoroughfare, with the precise distribution of industrial land, ware-housing land, and construction land.

Point 13: Page 7, Line 215-216, please add details about your investigation on the probability of being hit by typhoon. What about previous events in the study area? What about their impact on the local population, on buildings and infrastructure?

 [Response]: Thank you for your comments. We are sorry for the ambiguity in the previous expression. This view was based on existing literature and project experience. We have rewritten this section. The new sentence is shown below.

  • 4.1 Study Area Overview (Page 7, Line 247-253): Through the analysis of typhoons in the East China Sea over the years, existing research recognized that the latitude of the typhoon landfall point varies greatly, with a tendency to move northward [1]. Therefore, the probability of the Lin-gang Special Area being hit by a typhoon will increase. We need to pay attention to this phenomenon. In 2021, under the influence of the typhoon, the Lin-gang Special Area activated emergency plans to transfer susceptible groups and suspend some public transport many times [2,3].

References

  1. Lu, X.; Dong, C.; Li, G. Variations of typhoon frequency and landfall position in East China Sea from 1951 to 2015. Transactions of Atmospheric Sciences 2018, 41, 433-440.
  2. Available online: http://sh.cma.gov.cn/sh/news/qxyw/202109/t20210913_3813856.html (accessed on 3 August).
  3. Available online: http://sh.cma.gov.cn/sh/news/qxyw/202107/t20210724_3562269.html (accessed on 3 August).

Point 14: Page 7, Line 217-218, on what data is this view based? If it is based on existing literature, you should cite it in the text with appropriate citations and references in the respective list. If it is your own assessment, you should indicate what data you have based it on, so that we can determine whether this assessment is acceptable or not.

[Response]: Thank you for your comments. This view was also based on existing literature and project experience. We have rewritten this section and cited them in the text. The new sentence is shown below.

  • 4.1 Study Area Overview (Page 7, Line 254-259): The topographic analysis shows that the elevation is basically around 4m or less, and Shanghai's geological formation primarily comprises soft soil [1]. It has been established that excessive groundwater pumping, the construction of facilities and amenities, and high-rise buildings are the root causes of the land subsidence in Shanghai [2-4].

References

  1. Lyu, H.M.; Shen, S.L.; Zhou, A.; Yang, J. Risk assessment of mega-city infrastructures related to land subsidence using improved trapezoidal FAHP. Science of The Total Environment 2019, 717, 135310.
  2. Ye-Shuang, X.; Shui-Long, S.; Dong-Jie, R.; Huai-Na, W. Analysis of Factors in Land Subsidence in Shanghai: A View Based on a Strategic Environmental Assessment. Sustainability 2016, 8, 573.
  3. Wu, H.-N.; Shen, S.-L.; Yang, J. Identification of Tunnel Settlement Caused by Land Subsidence in Soft Deposit of Shanghai. Journal of Performance of Constructed Facilities 2017, 31, doi:10.1061/(asce)cf.1943-5509.0001082.
  4. Zhao, Y.; Zhou, L.; Wang, C.; Li, J.; Qin, J.; Sheng, H.; Huang, L.; Li, X. Analysis of the Spatial and Temporal Evolution of Land Subsidence in Wuhan, China from 2017 to 2021. Remote Sensing 2022, 14, doi:10.3390/rs14133142.

Point 15: Page 7, Line 220-228, I suggest to add citations on studies for the planning and the design of the study area.

[Response]: Thank you so much for your valuable suggestions. We agreed that it is necessary to add citations on studies for the study area's planning and design. So we added citations where appropriate. Citation 1 showed the overview of the Lin-gang Special Area, and citation 2 demonstrated a more detailed industrial plan. The new sentence is shown below.

  • 4.1 Study Area Overview (Page 7, Line 260-262): The planning of the Lin-gang Special Area is based on two crucial time points, 2025 and 2035 [1]. Industrial development is mainly based on high-tech industries, including hydrogen energy, biomedicine, integrated circuit, etc [2].

Citations

  1. Overview, Lin-gang Special Area. Available online: https://en.lgxc.gov.cn/2021-08/09/c_430834.htm (accessed on 2 August).
  2. Notice of the State Council on printing and distributing the overall plan for Lin-gang Special Area of the China (Shanghai) Pilot Free Trade Zone. Available online: https://en.lgxc.gov.cn/2019-12/24/c_463204.htm (accessed on 2 August).

Point 16: Figure 4, the image is of low analysis. Please replace with a figure with higher analysis. Please use different colors in different steps of the flowchart.

[Response]: Thank you for your comments. Figure 4 shows the intermediate process of sorting out the safety risk factors in the Lin-gang Special Area based on the causative factors in Figure 1, which is a picture of process nature. Therefore, this image was removed for the sake of conciseness of the article and has no remaining impact on the whole article.

Point 17: Page 7, Line 241-244, I suggest to add definitions for 1, 2, 3 and 4.

[Response]: Thank you for your comments. We have added definitions for 1, 2, 3 and 4 in manuscript. The new sentence is shown below.

  • 4.2.1 Identification and Categorization (Page 8, Line 278-285):
  1. Typhoon (S1R1): Typhoon is a cyclone with high wind and heavy precipitation, causing casualties and damage [1].
  2. Rainstorm and flood (S1R2): Rainstorms and floods often occur in the Yangtze River; intensive rainfall is a key factor in increasing flood hazards [2]
  3. Storm surge (S1R3): Storm surge is an increase in sea level caused by low atmospheric pressure and strong winds [3].
  4. Land subsidence (S1R4): Land subsidence, vertical land movements, includes natural and human-induced subsidence in sedimentary coastal lowlands [4].

References

  1. Johnson, K.; Depietri, Y.; Breil, M. Multi-hazard risk assessment of two Hong Kong districts. International Journal of Disaster Risk Reduction 2016, 19, 311-323, doi:10.1016/j.ijdrr.2016.08.023.
  2. Chan, F.; Yang, L.E.; Scheffran, J.; Mitchell, G.; Mcdonald, A. Urban flood risks and emerging challenges in a Chinese delta: The case of the Pearl River Delta. Environmental Science & Policy 2021, 122, 101-115.
  3. Muis, S.; Verlaan, M.; Winsemius, H.C.; Aerts, J.C.J.H.; Ward, P.J. A global reanalysis of storm surges and extreme sea levels. Nature Communications 2016, 7, 11969.
  4. Nicholls, R.J.; Lincke, D.; Hinkel, J.; Brown, S.; Fang, J. A global analysis of subsidence, relative sea-level change and coastal flood exposure. Nature Climate Change 2021.

Point 18: Figure 5, the image is of low analysis. Please replace with a figure with higher analysis. Please use different colors in different levels of the hierarchical structure. Please add more details in your figure captions throughout the manuscript.

[Response]: Thank you for your comments. We have replaced it with a figure with higher analysis (Page 10, Line 249). Now, it is named Figure 4 because the original Figure 4 has been deleted for brevity. And we have added more details in the figure captions. The new figure captions are shown below.

Figure 1. The triangular framework for security risk in new towns (Page 3, Line 120).

Figure 2. The framework of DEMATEL-ISM-SNA method (Page 5, Line 199).

Figure 3. Study area (Page 7, Line 245).

Figure 4. The hierarchical structure of the risk factors in the Lin-gang Special Area (Page 10, Line 349).

Figure 5. Transmission map of the relationship among safety risk factors in the Lin-gang Special Area (Page 12, Line 380).

Point 19: Table 4, I suggest to use bold letters for "Table 4".

[Response]: Thank you for your detailed review. We have used bold letters for "Table 2." (the original table 1 and 2 have been put in the appendix for brevity).

Point 20: Figure 6, I suggest to reverse the word "out-degree" in the caption of the vertical axis.

[Response]: Thank you so much for your valuable suggestions. As shown in the picture below, we have reversed the word "out-degree" in the caption of the vertical axis (Page 12, Line 380).

Figure 5. Transmission map of the relationship among safety risk factors in the Lin-gang Special Area.

Point 21: Page 12, Line 339-341, (i) directly influencing factors, (ii) intermediate influencing factors, and (iii) essential influencing factors.

[Response]: Thank you for your comments. In the new manuscript, we have replaced “directly influencing factors, intermediate influencing factors, and essential influencing factors” with “(i) directly influencing factors, (ii) intermediate influencing factors, and (iii) essential influencing factors” (Page 12, Line 383-385).

Point 22: Page 12, Line 346, I suggest to insert a new paragraph.

[Response]: Thank you for your detailed review. We have divided this part into a separate paragraph to make it clearer to read (Page 12, Line 391).

Point 23: Page 12, Line 348, I suggest to insert a new paragraph.

[Response]: Thank you for your detailed review. We have divided this part into a separate paragraph to make it clearer to read (Page 12, Line 395).

Point 24: Page 12, Line 346, “ :”

[Response]: Thank you for your detailed review. We have replaced “.” with “:” in the new manuscript (Page 13, Line 440).

Round 2

Reviewer 2 Report

No further comment.

Author Response

Manuscript Number: ijerph-1825009

Title: Hierarchical Structure Model of Safety Risk Factors in New Coastal Towns: A Systematic Analysis Using DEMATEL-ISM-SNA Method

Dear Editors and Reviewers,

We are grateful for the precious opportunity to revise our manuscript entitled "Hierarchical Structure Model of Safety Risk Factors in New Coastal Towns: A Systematic Analysis Using DEMATEL-ISM-SNA Method" (Manuscript ID: ijerph-1825009).

According to the reviewer's comments, we have extensively revised the language throughout the article to provide a better reading experience. First, we have fixed the grammar errors in the writing (Page 4, Line 160; Page 11, Line 350; Page 13, Line 424). Second, we have standardized the terminology in the manuscript, such as replacing "security risk" with "safety risk" (Page 2, Line 79; 84; 95 etc.) and adjusting Figure 1 (Page 3, Line 120) accordingly. Third, we have changed several sentences to make them more concise (Page 4, Line 138-139; 154; Page 7, Line 242; 260; Page 12, Line 405-406; Page 13, Line 454-455 etc.). Furthermore, we have modified the format of Table 2 (Page 11, Line 352) to align the data to the decimal point, making it more standardized.

Thank you again for your advice, attention, and timely response. We are looking forward to hearing from you at your earliest convenience.

Best regards,

The authors
